# Psychometric Performance of the Stony Brook Scar Evaluation Scale and SCAR-Q Questionnaire in Dutch Children after Pediatric Surgery

**DOI:** 10.3390/ijerph21010057

**Published:** 2023-12-30

**Authors:** Chantal A. Ten Kate, Hilde J. H. Koese, M. Jenda Hop, André B. Rietman, René M. H. Wijnen, Marijn J. Vermeulen, Claudia M. G. Keyzer-Dekker

**Affiliations:** 1Department of Pediatric Surgery and Intensive Care Children, Erasmus MC Sophia Children’s Hospital, Wytemaweg 80, 3015 CD Rotterdam, The Netherlands; c.tenkate@erasmusmc.nl (C.A.T.K.); hildekoese@gmail.com (H.J.H.K.); a.rietman@erasmusmc.nl (A.B.R.); r.wijnen@erasmusmc.nl (R.M.H.W.); 2Department of Plastic and Reconstructive Surgery, Erasmus University Medical Center, 3015 CD Rotterdam, The Netherlands; m.hop@erasmusmc.nl; 3Department of Child and Adolescent Psychiatry/Psychology, Erasmus MC Sophia Children’s Hospital, 3015 CD Rotterdam, The Netherlands; 4Department of Pediatrics, Division of Neonatology, Erasmus MC Sophia Children’s Hospital, 3015 CD Rotterdam, The Netherlands; m.j.vermeulen@erasmusmc.nl

**Keywords:** patient-reported outcome measures, psychometrics, postoperative, scars, surveys and questionnaires

## Abstract

*Introduction:* The growing population of survivors following pediatric surgery emphasizes the importance of long-term follow-up. The impact of surgical scars on daily life can be evaluated through patient-reported outcome measurements. The Stony Brook Scar Evaluation Scale (SBSES) and SCAR-Q questionnaire are two interesting instruments for this purpose. We evaluated their psychometric performance in Dutch children after pediatric surgery. *Methods:* After English–Dutch translation, we evaluated—following the COSMIN guidelines—the feasibility, reliability (internal and external), and validity (construct, criterion, and convergent) of the SBSES and SCAR-Q in Dutch patients < 18 years old with surgical scars. *Results:* Three independent observers completed the SB for 100 children (58% boys, median age 7.3 (IQR 2.5–12.1) years) in whom surgery had been performed a median of 2.8 (0.5–7.9) years ago. Forty-six of these children (61% boys, median age 12.1 (9.3–16.2) years) completed the SCAR-Q. Feasibility and internal reliability (Cronbach’s alpha > 0.7) was good for both instruments. For the SB, external reliability was poor to moderate (interobserver variability: ICC 0.46–0.56; intraobserver variability: ICC 0.74). For the SCAR-Q, external reliability was good (test–retest agreement: ICC 0.79–0.93). Validity tests (construct, criterion, and convergent) showed poor to moderate results for both instruments. *Conclusions:* The Dutch-translated SBSES and SCAR-Q showed good feasibility and internal reliability. External reliability and validity were likely affected by differences in conceptual content between the questionnaires. Combining them would provide insight in the impact of scars on patients. Implementation of these instruments in longitudinal follow-up programs could provide new insights into the long-term psychological outcome after pediatric surgery.

## 1. Introduction

Improved surgical techniques and intensive care treatment have resulted in a growing population of survivors after pediatric surgery [1]. Consequently, long-term outcomes such as quality of life have become more important, which can be negatively affected by psychosocial and physical problems [2]. More specifically, surgical scars can impose a burden in daily life, encompassing not only physical symptoms such as itchiness, pain, or numbness but also psychosocial and emotional problems stemming from the scar’s appearance [3,4]. Hiding scars, avoiding social interaction or leisure activities such as swimming, and struggling with emotional distress and anxiety have been documented in studies involving adults [4]. 

The available validated instruments to measure a child’s health status and perception of potential limitations [5,6], are generic tools that are inadequate to assess the specific aspects and experienced impact of scars on a child’s life. Validated tools for assessing surgical scars in children in the Netherlands are scarce. Although the Patient and Observer Scar Assessment Scale (POSAS)—which evaluates the visual, tactile, and sensory characteristics of scars—has been validated in the Dutch language as an observer-report for surgical scars in all ages groups, as a patient-report it is only suitable for adults [7]. Moreover, it does not address the psychological impact of scarring and is unsuitable for photographic evaluation.

Searching for alternative instruments, we found two that stood out. One is the SCAR-Q, which measures the aesthetic appearance, symptoms, and psychosocial impact of both burn and surgical scars, and is developed for patients aged ≥8 years old [8]. The other is the Stony Brook Scar Evaluation Scale (SBSES), which measures the aesthetic appearance of surgical scars and is developed for photographic observer evaluation [9], making it a pragmatic tool usable at all ages and in various settings. Both instruments have not yet been validated in the Netherlands, however.

The implementation of patient-reported outcome measures enhances the communication between patients and professionals [10]. It can therefore be expected that standardized scar evaluation during post-operative follow-up would benefit patient care. Addressing the physical and psychosocial impact in combination with an objective measurement would provide a complete overview of the impact of scars on a patient’s life. Intending to implement these instruments in our longitudinal follow-up program, both instruments first need to be validated in the Dutch language. In this validation study, we assessed the psychometric properties using the SCAR-Q and the SB in Dutch children with surgical scars [11]. 

## 2. Methods

This cross-sectional cohort study was conducted following the COSMIN guidelines [12]. The institutional review board approved this study (MEC-2021-0060).

### 2.1. Study Population

Participants were eligible if they had sufficient knowledge of the Dutch language, were <18 years old, and had a previous history of laparotomy, thoracotomy, or excision of soft tissue lesions of the extremities. Children with known intellectual disability were excluded. 

### 2.2. Data Collection

Data were collected between February and June 2021 at the outpatient clinic of the department of Pediatric Surgery of a tertiary academic hospital. After written (parental) informed consent, a member of the research team (H.K.) completed the POSAS on sight, and the scar was photographed according to a standardized protocol (see Appendix A). Two pediatric surgeons (C.K.D., R.W.) and one plastic surgeon (J.H.) independently completed the SBSES for the anonymized photographs to evaluate the interobserver variability. J.H. completed the SBSES again six months later to evaluate the intraobserver variability. 

Children ≥ 8 years old were asked to complete the SCAR-Q on paper at the outpatient clinic. To examine the test–retest agreement, they were asked to complete the SCAR-Q again two weeks later on paper by post.

The following data were collected at the outpatient visit: skin type according to Fitzpatrick [13], and scar location, length, and width. Additional data were retrieved from the patient records: sex, date of birth, date and type of surgery, height and weight, and previous surgery on the same scar. 

### 2.3. Instruments

The SBSES [9] is a 5-item observer-report for photographic evaluation of linear scars: width, height, color, hatch/suture marks, and overall appearance. Each item is scored either 0 or 1. The total score is the sum score of the five items, varying between 0 (worst) and 5 (best).

The SCAR-Q [8] is a self-report questionnaire for children aged 8–17 years old. It encompasses three domains: appearance scale, symptom scale, and psychosocial impact. Each item is scored on a 4-point Likert scale from “doesn’t bother at all” to “bothers very much”. Total scores are calculated per domain by transforming the sum score, ranging from 0 (worst) to 100 (best).

The POSAS [7] is an observer-report for onsite evaluation by a physician of both linear and burn scars. It consists of seven items: vascularity, pigmentation, thickness, relief, pliability, surface area, and overall opinion. Each item is scored from 1 (normal skin) to 10 (worst scar imaginable). Additional descriptions can be added. The total score is the sum of the six item scores, ranging from 6 (best) to 60 (worst). 

More details on the instruments, and a standardized protocol for scar photography are included in the Appendix A.

### 2.4. Statistical Analysis

Sample size was set to 100 patients, based on the COSMIN criteria [12]. Data are presented as number (%) or median (interquartile range (IQR)). Individual items of the SBSES and SCAR-Q were described as mean ± SD (range). Feasibility was considered good if <10% of the items had >5% missing values [14]. Floor and ceiling effects (percentage of respondents reporting, respectively, the minimum and maximum possible score) were considered acceptable if occurring in <15% of the cases [15].

Internal reliability was considered poor, moderate, or good (Cronbach’s alpha, respectively, <0.5, 0.5–0.7, or >0.7) [15]. External reliability was tested through test–retest agreement for the SCAR-Q, and interobserver and intraobserver variability for the SBSES, using intraclass coefficients (ICCs)—both with a two-way mixed model, single measures, and absolute agreement—and considered poor (<0.50), moderate (0.50–0.74), good (0.75–0.90), or excellent (>0.90) [16]. 

Construct validity was examined through known-groups validity. We compared the SCAR-Q scores of patients with and without scars visible when wearing daily life clothes, using Mann–Whitney U-tests. The associated effect sizes (ESs) were calculated [17] and considered strengthening for the validity if moderate (>0.30) or large (>0.50). Children with visible scars were hypothesized to have lower scores. Due to the setup with photographic assessment, construct validity could not be examined for the SBSES scores. 

Criterion validity was evaluated through concurrent validity by correlating the SCAR-Q and SBSES scores with the size (length and width) of the scar, using Spearman’s rho (*r_s_*), and concluded as poor (<0.40), moderate (0.40–0.59), good (0.60–0.79), or excellent (>0.80) [15]. To correlate the individual—dichotomous—items with the size (length and width) of the scar, we used a point-biserial correlation (*r_pb_*). 

Convergent validity was assessed by correlating the SCAR-Q and SBSES scores with the—previously validated—POSAS scores, using Spearman’s rho (*r_s_*), and concluded as poor (<0.40), moderate (0.40–0.59), good (0.60–0.79), or excellent (>0.80) [15]. 

All statistical analyses were performed using SPSS V.25.0 (IBM, Chicago, IL, USA), with a significance level of *p* < 0.05.

## 3. Results

### 3.1. Study Population

In total, 140 children with linear scars visited the outpatient clinic during the study period, of whom 132 were eligible. One child’s scar was not visible due to a stoma plaque placed over it and could therefore not be evaluated. Eighteen others refused to participate, and 13 children could not participate due to logistic reasons. Thus, 100 children (76% of eligible, see Figure 1) with a median age of 7.3 (2.5–12.1) years old participated. All skin types were represented; 24% were of color (type IV–VI). Most children (75%) had undergone a laparotomy (n = 62 transverse incision, n = 13 midline incision). Overall, surgery had been performed a median of 2.8 (0.5–7.9) years ago. A subgroup of 46 children aged ≥8 years, median age of 12.1 (9.3–16.2) years old, completed the SCAR-Q. For them, surgery had been performed a median of 8.5 (2.2–12.0) years ago. Table 1 presents all patient characteristics.

### 3.2. Stony Brook Scar Evaluation Scale

#### 3.2.1. Item Evaluation, Internal Reliability, and External Reliability

There were no missing values, representing a good feasibility of the scale (Appendix A). Observers reported median total scores of, respectively, 4, 3, and 4 (mean total scores, respectively, 3.55 ± 1.34, 2.73 ± 1.54, and 3.40 ± 1.43). Internal reliability was good, with Cronbach’s alpha of 0.73, 0.75, and 0.74 for each of the observers. External reliability was poor to moderate (ICC ranging 0.46–0.56) for the interobserver variability and moderate (ICC 0.74) for the intraobserver variability. When comparing the individual items, the interobserver variability was slightly better, though all ICCs remained at <0.75 (Appendix A).

#### 3.2.2. Criterion and Convergent Validity

Total SBSES scores correlated poorly with the scar length (*r_s_* ranging −0.31–0.21) and poorly to moderately with the scar width for all observers (*r_s_* ranging −0.49–0.36, criterion validity; Table 2). When comparing the individual items scored per observer, we found a significant correlation for the scar length with the items “width”, “height”, and “overall score” for observer 3 (*p*-value, respectively, 0.04, 0.002, and 0.04, criterion validity; Appendix A). For the scar width, we found a significant correlation with the items “width” and “overall score” for all observers (*p*-value ranging from 0.001–0.04, criterion validity; Appendix A). 

Total SBSES scores showed a poor to moderate correlation with total POSAS scores ((*r_s_* ranging −0.5–0.36) convergent validity, Table 2). When comparing two semicomparable items of the SBSES and the POSAS (respectively, color versus vascularity and height versus thickness), correlations were poor as well (Appendix A).

### 3.3. SCAR-Q

#### 3.3.1. Item Evaluation, Internal Reliability, and External Reliability

Feasibility of the SCAR-Q was good, with only one missing item in the appearance scale for one child (Appendix A). Total scores for each domain are presented in Table 3. Any significant floor effects were absent. Ceiling effects of >15% were present for all domains. Internal reliability was good, with Cronbach’s alpha, respectively, 0.94, 0.81, and 0.91 per domain. Twenty-nine children (63.0% of the original sample) returned the retest. Basic characteristics did not differ between respondents and nonrespondents. Test–retest agreement was good for the appearance and psychosocial scale (ICC, respectively, 0.85 and 0.79), and excellent for the symptom scale (ICC 0.93). 

#### 3.3.2. Construct, Criterion, and Convergent Validity

Children with scars visible when wearing daily life clothes did not have significantly lower SCAR-Q scores than those with scars hidden (construct validity; Table 3). Total SCAR-Q scores correlated poorly with the scar length and width for all domains (criterion validity; Table 3). Total SCAR-Q scores were poorly correlated with total POSAS scores ((*r_s_* ranging −0.18–0.07); convergent validity, Table 3). When comparing the individual items, correlations were poor as well (Appendix A).

## 4. Discussion

In this validation study of two condition-specific instruments, we evaluated the psychometric performance of the Dutch-translated SBSES and SCAR-Q in children with surgical scars. Implementation of these tools in clinical practice could potentially benefit patient care and long-term follow-up. Both instruments showed a good feasibility. For the SBSES, internal reliability was good, but external reliability was poor to moderate. For the SCAR-Q, internal and external reliability were good to excellent. However, validity tests (construct, criterion, and convergent) showed poor to moderate results for both instruments. 

While the SBSES has been used in numerous studies worldwide, it had not been previously validated in any language other than English. In our study, the psychometric performance was slightly lower compared to the original study [9], for example, in terms of interobserver variability and convergent validity. Comparing our SBSES scores with those from other pediatric studies is challenging since they either excluded linear scars [18,19] or included revision surgery [20]. When comparing with two recent adult studies, including post-cesarean and post-surgical facial scars, our findings indicated lower scores (mean SBSES scores ranged from 2.73–3.55, compared to 2.79–3.49 and 4.2–4.5, respectively) [21,22]. Notably, the latter study involved patients with a mean age of 70.6 years old. A patient’s age could potentially influence an observer’s score, but, due to the insufficient data, this remains highly speculative and necessitates further investigation.

The SCAR-Q has been translated in twelve languages, and validated in the French, Arabic, and Italian language [23,24,25]. The French and Arabic study investigated adults. The Arabic study published data on cognitive debriefing, not on validation in the field. Mean total scores in our population were slightly lower than those reported in the French population (appearance, symptom, and psychosocial scale, respectively, 73.6, 81.3, and 85.3) [23]. Other reliability and validity results were similar. The Italian research team published their results as a letter to the Editor, unclear about the age of the participants and lacking any results in details. Specific pediatric studies remain unavailable.

One possible explanation why in our study scores for both instruments were overall lower than those reported before is that we only included surgical scars, whereas others included traumatic and burn scars as well [23,26]. The possibility of illusionary bias should also be considered, wherein a person might incorrectly asses a particular aspect of their condition due to coping mechanisms [27]. It remains uncertain, however, whether our data reflect underestimation or a generally elevated level of acceptance, because data on psychosocial and peer support were not available. 

Third, the time elapsed since surgery is of great impact. In 35% of our patients, surgery had been performed less than a year ago, whereas in the original study, 36% of the scars were >5 years old, and in the French study, 75% of the scars were >1 year old. The last stage of wound repair—remodeling—begins 2–3 weeks after injury, and can last for over a year [28]. In alignment with this, within our population, patients who had undergone surgery <1 year ago reported lower SCAR-Q scores than those who had undergone surgery ≥1 year ago (Appendix A). 

Last, regarding skin composition, children’s epidermis and stratum corneum are thinner than those of adults. Moreover, infant skin contains more water and fewer natural moisturizing factors. On a cellular level and regarding microstructure, children’s and adult’s skins differ as well [29]. These factors could affect scar healing, and thereby explain the lower SB and SCAR-Q scores in children compared to adults.

When examining the psychological performance in our study, certain aspects deserve attention. First, ceiling effects were noted for the SCAR-Q. This commonly observed phenomenon in PROMs is most likely related to the naturally right-skewed quality of life experienced by patients [5,30,31]. 

Second, the external reliability of the SBSES was less satisfying than expected. The difference between the three observers—who varied in field of expertise and years of experience—reminds us to remain careful when interpreting the scores in clinical practice. Conclusions cannot be drawn from one individual’s judgement. Nonetheless, the stronger intraobserver variability with a six-month interval confirms the reliability of the instrument and its value for longitudinal evaluation. 

Third, the validity results of both instruments require critical appraisal. The different approaches to determine whether an instrument “truly measures the construct it purports to” are a well-established topic of discussion [12]. In this study, we used as a gold standard the POSAS—a validated and broadly used instrument worldwide. Considering that a higher POSAS score reflects a worse outcome, our study shows a moderate correlation with the SBSES but a poor correlation with the SCAR-Q. This is not surprising, though, since the POSAS is an observer-report which does not reflect the feelings a child may have about their scar. It is important to acknowledge the difference between observer-reports and self-reports, along with the fact that the SBSES and SCAR-Q measure two different concepts that complement each other. 

Remarkably, in the present study, scar size did not correlate well with the SBSES, though width and height are two of the five items of this instrument. This lack of correlation may be subject to observer bias, a phenomenon previously described in studies with medical images [32]. Our observers found it difficult to score a photograph of a fresh scar, because they know it will still improve aesthetically. To illustrate, 9% of the scars were from surgeries performed less than one month ago. Adequate training of observers could potentially prevent this bias.

To the best of our knowledge, this study represents the initial validation of both the SBSES and SCAR-Q in Dutch children, with a specific focus on surgical scars. Burn and traumatic scars differ in both aesthetic and symptomatic aspect, which could distort the results. The applied inclusion criteria and methods adhered to the COSMIN guidelines, and response rates were high. Since 24% of the participants had skin of color, our population was a representative sample mirroring the broader Dutch population [33]. Some limitations should be addressed. First, recruitment of participants from a tertiary hospital may have introduced a selection bias, favoring patients with a more severe medical history. The setup of a single center study could have contributed to a potential selection bias as well. Second, we did not collect data from nonparticipants; thereby, selection bias cannot be ruled out. Third, a considerable portion of the examined scars had not undergone complete remodeling, which might have influenced the accuracy of the outcomes. Following from this, it would be interesting to investigate how a scar develops in a young child versus an older child. Moreover, additional postoperative therapies—though not present in our study population—such as radiotherapy or use of steroids could influence the development of a scar as well. However, this was beyond the scope of this particular validation study. Last, though it is recognized that females report lower quality of life than do males, investigating sex-specific SCAR-Q scores was beyond the scope of our study. 

Overall, the SBSES and SCAR-Q can both be considered suitable instruments for research purposes and clinical practice. However, as they measure two different concepts, combining them—perhaps even with the POSAS—would provide a holistic overview of the impact of a surgical scar on a child. Considering that the SBSES allows for photographic evaluation, fewer hospital visits could be required for the longitudinal follow-up program. However, photographs taken at home could have insufficient quality, which may affect the reproducibility of our study results. The addition of the SCAR-Q—which could also be completed from home—would provide unique insights in the psychosocial outcome of these children. 

Still, in this age of value-based healthcare [34], where it is important to prioritize treatment in relation to what matters most to a patient, a hospital visit can be beneficial. A physician’s judgement and treatment plan is influenced by their observation of the veritable corporeal and emotional burden a child experiences. Moreover, the child having completed the SCAR-Q prior to a consultation could provide the opportunity to discuss delicate issues. 

Based on clinical experience, we suggest to complete both instruments at six weeks post-operatively—to detect hypertrophic scars and start proper treatment to prevent aggravation; at one year post-operatively—to consider revision surgery; and then every five years until adulthood—to observe changes by growth. 

## 5. Conclusions

The Dutch-translated SBSES and SCAR-Q showed good feasibility and internal reliability. External reliability and validity were most likely affected by the differences in conceptual content between the questionnaires. Combining them would best reflect the impact of a surgical scar on a child. For future perspectives, implementation of these instruments in longitudinal follow-up programs may provide new insights into the long-term outcome after pediatric surgery and, when necessary, lead to adequate treatment of disturbed scars. 

## Figures and Tables

**Figure 1 ijerph-21-00057-f001:**
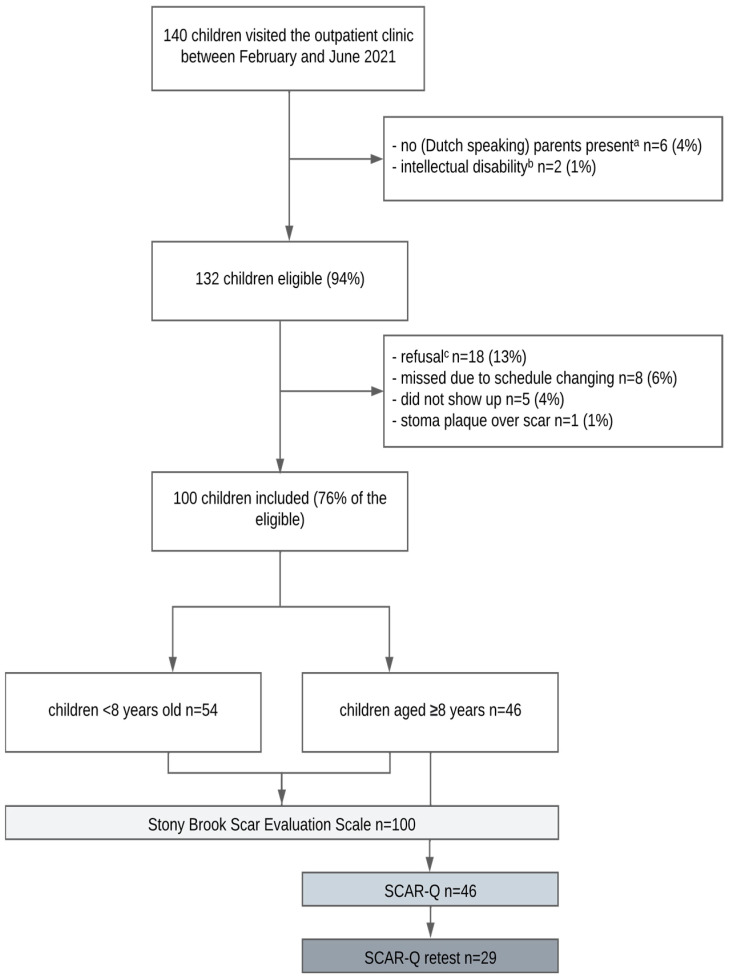
Flowchart of the study population. ^a^ Other family member present, e.g., brother, sister, or grandparents. ^b^ Autism spectrum disorder (n = 1), unspecified (n = 1). ^c^ Sleeping child (n = 2), other appointments (n = 15), child traumatized (n = 1).

**Table 1 ijerph-21-00057-t001:** General characteristics of the respondents, presented as n (%) or as median (interquartile range). SBSES = Stony Brook Scar Evaluation Scale. ^A^ According to Fitzpatrick types of skin scale [13]. ^B^ Stoma reversal n = 20 (after anorectal malformation (n = 7), Hirschsprung’s disease (n = 6), ileal atresia (n = 1), complicated appendectomy (n = 1), gastroschisis (n = 1), necrotizing enterocolitis (n = 1), ileus with unknown cause (n = 1), type 5 familial hemophagocytic lymphohistiocytosis (n = 1), lymphangiomatous malformation in the abdomen (n = 1)), congenital diaphragmatic hernia (n = 16), esophageal atresia (n = 11), congenital heart disease (n = 8), atresia of the duodenum, jejunum, or colon (n = 6), volvulus (n = 5), incisional hernia (n = 4, individual cases of gastroschisis, atresia of the jejunum, or ileum, congenital diaphragmatic hernia), ileus (n = 3, individual cases of congenital diaphragmatic hernia, anorectal malformation, primary ileus), bowel resection (n = 2), gastroschisis (n = 2), duplication cysts (n = 2 duodenum, stomach), omphalocele (n = 2), lipoma excision (n = 1), unspecified skin tumor excision (n = 1), pectus excavatum (n = 1), pectus carinatum (n = 1), and other individual cases (n = 15): cystic splenectomy, jejunal obstruction (after splenectomy and congenital diaphragmatic hernia surgery complication), segment esophageal resection after lye ingestion, fasciotomy, pancreas resection after abdominal trauma, necrotizing enterocolitis, malposition percutaneous endoscopic gastrostomy, lobectomy, kidney transplantation, intussusception, epigastric hernia, cystic hemi-hepatectomy, duodenal web, intestinal perforation, Hirschsprung’s disease, appendectomy.

	SBSES (n = 100)	SCAR-Q (n = 46)
Demographic characteristics		
Region		
North	1 (1.0)	1 (2.2)
East	7 (7.0)	3 (6.5)
South	25 (25.0)	11 (23.9)
West	67 (67.0)	31 (67.4)
Child characteristics		
Male	58 (58.0)	28 (60.9)
Age in years	7.0 (2.5–12.0)	12.1 (9.3–16.2)
Length (cm)	122.0 (86.3–148.0)	148.4 (133.9–164.3)
Weight (kg)	22.6 (11.8–36.9)	37.0 (30.2–51.6)
Skin type ^A^		
Type I—very fair	5 (5.0)	2 (4.3)
Type II—fair	56 (56.0)	24 (52.2)
Type III—bit tinted	15 (15.0)	8 (17.4)
Type IV—tinted	14 (14.0)	7 (15.2)
Type V—dark	4 (4.0)	3 (6.5)
Type VI—very dark	6 (6.0)	2 (4.3)
Scar characteristics		
Type of surgery ^B^		
Laparotomy	75 (75.0)	35 (76.1)
Thoracotomy	13 (13.0)	6 (13.0)
Sternotomy	9 (9.0)	4 (8.7)
Excision	3 (3.0)	1 (2.2)
Time since surgery (years)	2.8 (0.5–7.9)	8.5 (2.2–12.0)
Scar location		
Thorax	22 (22.0)	10 (21.7)
Abdomen	75 (75.0)	35 (76.1)
Upper extremities	2 (2.0)	0 (0.0)
Lower extremities	1 (1.0)	1 (2.2)
Scar length (cm)	8.6 (6.1–11.0)	10.5 (7.5–13.1)
Scar width (cm)	0.2 (0.1–0.3)	0.2 (0.1–0.4)
Previous surgery within same scar	22 (22.0)	9 (19.6)

**Table 2 ijerph-21-00057-t002:** Descriptive values of the Stony Brook Scar Evaluation Scale. Results of the Patient and Observers Scar Assessment Scale (POSAS) Observer scale can be found in Appendix A. ICC = intraclass correlation coefficient, CI = confidence interval.

**Descriptive values** **& internal reliability**				**Observer 1, Pediatric Surgeon (n)**	**Observer 2, Pediatric Surgeon (n)**	**Observer 3, Plastic Surgeon (n)**
	Width > 2 mm			32	65	40
	Elevated or depressed height in relation to surrounding skin			14	44	28
	Darker color than surrounding skin (red, purple, brown, or black)			37	39	31
	Suture marks present			33	36	21
	Overall appearance poor			29	43	40
	Total score (range)			4 (0–5)	3 (0–5)	4 (0–5)
	Cronbach’s alpha			0.731	0.748	0.742
**External reliability**			**Level of agreement, ICC (95% CI)**			
	Observer 1 vs. observer 2		0.48 (0.20–0.66)			
	Observer 1 vs. observer 3		0.56 (0.41–0.68)			
	Observer 2 vs. observer 3		0.46 (0.25–0.62)			
**Criterion validity**		**Level of agreement, *r_s_* (95% CI)**				
		Scar length	Scar width			
	Observer 1	−0.20 (−0.38–0.004)	−0.36 (−0.52–0.17)			
	Observer 2	−0.21 (−0.39–0.01)	−0.42 (−0.57–0.24)			
	Observer 3	−0.31 (−0.47–0.12)	−0.49 (−0.57–0.24)			
**Convergent validity**			**Level of agreement, *r_s_* (95% CI)**			
	POSAS vs. observer 1		−0.52 (−0.65–0.36)			
	POSAS vs. observer 2		−0.36 (−0.65–0.36)			
	POSAS vs. observer 3		−0.58 (−0.70–0.44)			

**Table 3 ijerph-21-00057-t003:** Descriptive values of the SCAR-Q questionnaire. Results of the Patient and Observers Scar Assessment Scale (POSAS) Observer scale can be found in Appendix A. IQR = interquartile.

**Descriptive values**		**Items**	**Respondents (n)**	**Median (IQR)**		**Mean ± SD**			**Floor, n (%)**	**Ceiling, n (%)**		**Cronbach’s Alpha**	
	Appearance	12	46	73.00 (59.75–100.00)		71.30 ± 21.45			-	12 (26.1)		0.94	
	Symptom	12	46	77.00 (63.00–100.00)		78.13 ± 16.41			-	12 (26.1)		0.81	
	Psychosocial	5	46	87.00 (67.50–100.00)		81.04 ± 23.85			1 (1.1)	20 (43.5)		0.91	
**External reliability**		**Respondents (n)**		**Level of agreement, ICC (95% CI)**									
	Appearance	29		0.85 (0.71–0.93)									
	Symptom	29		0.93 (0.67–0.92)									
	Psychosocial	29		0.79 (0.54–0.90)									
**Construct validity**				**Yes**			**No**						
				n	Median (IQR)		n	Median (IQR)			*p*-value		Effect size (r)
	Scars visible in daily life clothing												
	Psychosocial			5	64.00 (61.00–86.50)		41	73.00 (59.00–100.00)			0.758		−0.05
	Appearance			5	65.00 (62.00–91.00)		41	77.00 (65.00–100.00)			0.631		−0.08
	Symptom			5	100.00 (78.00–100.00)		41	87.00 (63.00–100.00)			0.370		−0.14
**Criterion validity**		**Level of agreement, *r_s_* (95% CI)**											
		Scar length		Scar width									
	Appearance	0.26 (0.03–0.05)		−0.11 (-0.39–0.18)									
	Symptom	0.14 (−0.15–0.42)		0.07 (-0.23–0.35)									
	Psychosocial	0.13 (−0.17–0.40)		−0.12 (-0.39–0.18)									
**Convergent validity**				**Level of agreement, *r_s_* (95% CI)**									
	POSAS vs. appearance			−0.12 (-0.39–0.18)									
	POSAS vs. symptom			−0.07 (-0.35–0.23)									
	POSAS vs. psychosocial			−0.18 (-0.45–0.11)									

## Data Availability

All dataset(s) supporting the conclusions of this article are available upon reasonable request.

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
