# Peer review of "Psychometric Performance of the Stony Brook Scar Evaluation Scale and SCAR-Q Questionnaire in Dutch Children after Pediatric Surgery"

_ijerph, 2023, doi:10.3390/ijerph21010057_

Round 1
Reviewer 1 Report
Comments and Suggestions for Authors
Interesting paper about the scar evaluation tools after surgery in pediatric population.
Given the nature of the cross sectional study, would a larger sample be more reliable to draw conclusion?
The statistical analysis used is not common to follow and I believe that more simple tables could help the readers get the message
Author Response
Reviewer 1
Interesting paper about the scar evaluation tools after surgery in pediatric population.
Given the nature of the cross sectional study, would a larger sample be more reliable to draw conclusion?
Thank you for reading our manuscript. Following the COSMIN guidelines, we based our sample size on the COSMIN Study Design checklist for Patient-reported outcome measurement instruments (Mokkink, 2019). This checklist states that 100 patients is an appropriate number to analyze the structural validity “adequate” or even “very good”. If one aims to perform a Rasch model or item response theory model, this number has to be increase to >200, or even to >1000 for an appropriate Mokken scale analysis. However, since the aim of this study was to assess the validity, reliability and feasibility of these two instruments, we deliberately set the sample size to n=100 to lessen the burden for the patients.
The statistical analysis used is not common to follow and I believe that more simple tables could help the readers get the message.
We acknowledge that the statistical analysis of psychometric properties is not common knowledge to every reader. We based our methods on prior experiences and studies in this field (Dellenmark-Blom et al., Journal of Pediatric Gastroenterology and Nutrition, 2018; ten Kate et al., Children, 2022). We have tried to make the analysis easier to follow by keeping the same order in the Methods section, Results section and Tables. We agree with the reviewer that the tables might be a bit unconventional because of their size. A potential alternative would be to divide them into separate tables. Since this would lead to a total of 10 tables, we leave this decision to the editor.
Reviewer 2 Report
Comments and Suggestions for Authors
This prospective cohort study examined the psychological impact of childhood surgical scarring on the individual and (possibly) family members as assessed by the SCAR-Q and SBSES scales. The SBSES, by analogy to the SCAR-Q, is presumed to reflect the psychological state of those around them (in most cases, their parents). I appreciate that this is a valuable report, given the scarcity of similar clinical studies. Therefore, I expect you to promptly correct the following two points and submit the report again.
Problem 1: In the Results section, it is stated that the total SBSES score has a low correlation with the length and width of the scar. Identifying items with high correlations would be important information for the treatment plan from the viewpoint of developing countermeasures. Therefore, please also analyze the other items of the SBSES, color, hatch/suture marks, and overall appearance, and include the results in Table 3 or in the text. Since these items are qualitative, unlike the width and length of the surgical scar, the analysis method can be either Chi-square or multivariate.
Problem 2: Please cite any reports that examine SBSES and the psychological state of the parents and add them to the discussion.
Author Response
Reviewer 2
This prospective cohort study examined the psychological impact of childhood surgical scarring on the individual and (possibly) family members as assessed by the SCAR-Q and SBSES scales. The SBSES, by analogy to the SCAR-Q, is presumed to reflect the psychological state of those around them (in most cases, their parents). I appreciate that this is a valuable report, given the scarcity of similar clinical studies. Therefore, I expect you to promptly correct the following two points and submit the report again.
Problem 1: In the Results section, it is stated that the total SBSES score has a low correlation with the length and width of the scar. Identifying items with high correlations would be important information for the treatment plan from the viewpoint of developing countermeasures. Therefore, please also analyze the other items of the SBSES, color, hatch/suture marks, and overall appearance, and include the results in Table 3 or in the text. Since these items are qualitative, unlike the width and length of the surgical scar, the analysis method can be either Chi-square or multivariate.
Thank you for your positive review of our manuscript. We have analyzed the correlation between the length and width of the scar with the individual SBSES items in Supplementary Table S3.3. We agree with you that the method we had initially chosen was not representing the genuine correlation optimally. Since the width and length of the scar are continuous variables, and the individual SBSES items are dichotomous variables, we have reran all analyses with a point-biserial correlation. We have added this to the Methods section (line 91-92) and the Results section (line 126-130), and have adjusted Supplementary Table S3.3.
Problem 2: Please cite any reports that examine SBSES and the psychological state of the parents and add them to the discussion.
The Stony Brook Scar Evaluation Scale is an instrument to assess the long-term aesthetic or cosmetic appearance of scars, which correlates with other previously validated cosmetic scales (Singer et al., Plastic and Reconstructive Surgery, 2007). It is an observer-report by physicians, which has never been described together with the psychological state of the parents in previous literature. We have discussed the most relevant studies that examine the SBSES in the second paragraph of the Discussion section, line 163-171.
Reviewer 3 Report
Comments and Suggestions for Authors
A remarkable study examining scar scoring in the pediatric age group.
Introduction-purpose: the purpose of the study should be stated more clearly. Is this study a validation study? If not, are the aforementioned scoring systems validated for the Dutch language? This information should be given in more detail and clearly.
The flowchart and final patient number details, including the excluded patients, presented in the Results section, should be moved to the materials-methods section.
A current study on the subject is "Salzillo R, Barone M, Persichetti P. Does a High-Quality Scar Overcome its Length? Italian Validation of the SCAR-Q Questionnaire. Aesthetic Plast Surg. 2023 Oct;47(5):2209- 2210. doi: 10.1007/s00266-023-03406-y. Epub 2023 May 30. PMID: 37253844." should be examined and discussed
Discussion-conclusion: The conclusion, especially the last two sentences, cannot be directly supported by data. Conclusion should be organized on this axis.
Author Response
Reviewer 3
A remarkable study examining scar scoring in the pediatric age group.
Introduction-purpose: the purpose of the study should be stated more clearly. Is this study a validation study? If not, are the aforementioned scoring systems validated for the Dutch language? This information should be given in more detail and clearly.
Thank you for the careful appraisal of our manuscript. We appreciate you pointing out that the purpose of our study is not clear to the reader. Indeed, this is a validation study, since both the SBSES and the SCAR-Q have not been validated in the Dutch language. We have clarified this in on page 3, line 29-31: “Intending to implement these instruments in our longitudinal follow-up program, both instruments first need to be validated in the Dutch language. In this validation study, we assessed the psychometric properties the SCAR-Q and the SB in Dutch children with surgical scars[11].”
The flowchart and final patient number details, including the excluded patients, presented in the Results section, should be moved to the materials-methods section.
We acknowledge how one can differ in their opinion about the set-up of a manuscript. We deliberately chose to use the Methods section to describe in detail on which criteria we selected our patient population, in order to allow reader to replicate our steps. We have included the patient numbers in our Results section, since these describe the result of our selection criteria. We hereby adhered to the Instruction for Authors on the website of IJERPH, and the STROBE (StrengThening the Reporting of OBservational studies in Epidemiology) checklist.
A current study on the subject is "Salzillo R, Barone M, Persichetti P. Does a High-Quality Scar Overcome its Length? Italian Validation of the SCAR-Q Questionnaire. Aesthetic Plast Surg. 2023 Oct;47(5):2209- 2210. doi: 10.1007/s00266-023-03406-y. Epub 2023 May 30. PMID: 37253844." should be examined and discussed.
We apologize for the omission of this recently published paper. We have discussed this study in the Discussion section, line 172-178, and have added the paper to the References.
Discussion-conclusion: The conclusion, especially the last two sentences, cannot be directly supported by data. Conclusion should be organized on this axis.
We agree with the reviewer that the conclusion should be supported by the information described in the paper. We have drawn our conclusion from all sections of the manuscript. To clarify our intents, we have changed the last sentence on page 10, line 257-260: “For future perspectives, implementation of these instruments in longitudinal follow-up programs may provide new insights in the long-term outcome after pediatric surgery, and when necessary lead to adequate treatment of disturbed scars.”
Reviewer 4 Report
Comments and Suggestions for Authors
Summary:
Manuscript ijerph-2723208 describes a clinical research paper evaluating the performance of two different questionnaires / scoring systems to evaluate the long-term outcomes of pediatric patients who underwent general surgical procedures as far as overall scar appearance, functional aspect and psychosocial impact. The authors translated the Stoney Brook Scar Evaluation Scale as well as SCAR-Q. They showed that these evaluations and reliable. There is however some significant inter-individual variability, however combination of different tests may allow longitudinal evaluations of functional and psychological impact of scars in children.
Comments:
Study design:
The study is well designed and uses questionnaires that have been mostly validated in the adult population. Having only three surgeons evaluate the scars may cause some methodological problems and could introduce bias or make it underpowered, as shown as the limited consistency of their responses. Please comment. Was there a reason why the study was conducted during covid? Did the authors take the time of the surgical intervention into account in their evaluation, versus overall chronological age?
Material and methods. This section is well described.
Results:
This is section is well detailed. The authors may want to include some further statistical data into Table 1, to evaluate for possible differences between the groups.
Introduction and discussion: both sections are well written and documented. The authors may consider to discuss some of the limitation of a single center study and also the impact of other treatment onto scar formation and acceptance. Some patients having suffered from solid tumors and who are long term survivors may not be bothered as much by a scar compared to patients with benign disease? Could the authors comment on that. Radiation fields, other medical therapies, especially steroids may worsen healing processes and make scar look worse. Please comment.
Minor comment:
Some spelling errors and minor grammatical inaccuracies should be corrected. A few medical terms should be translated into English, i.e. hernia cicatricalis is likely incisional hernia, necrotic enterocolitis is usually described as necrotizing enterocolitis in the English literature.
Comments on the Quality of English Language
The article is well-written, only minor grammatical errors or spelling errors should be corrected. Please see above.
Author Response
Reviewer 4
Manuscript ijerph-2723208 describes a clinical research paper evaluating the performance of two different questionnaires / scoring systems to evaluate the long-term outcomes of pediatric patients who underwent general surgical procedures as far as overall scar appearance, functional aspect and psychosocial impact. The authors translated the Stoney Brook Scar Evaluation Scale as well as SCAR-Q. They showed that these evaluations and reliable. There is however some significant inter-individual variability, however combination of different tests may allow longitudinal evaluations of functional and psychological impact of scars in children.
Comments:
Study design:
The study is well designed and uses questionnaires that have been mostly validated in the adult population. Having only three surgeons evaluate the scars may cause some methodological problems and could introduce bias or make it underpowered, as shown as the limited consistency of their responses. Please comment. Was there a reason why the study was conducted during covid?
Thank you for your detailed review of our manuscript. Our data was collected between February-June 2021. It was worldwide unclear how long the COVID pandemic would last. In our hospital, we have always continued all research projects as much as possible. However, at the time of our inclusions, restrictions in The Netherlands were very minimally. We therefore do not think that the COVID pandemic was of influence of our results, and therefore also have not included this as a potential bias in the Discussion section.
Did the authors take the time of the surgical intervention into account in their evaluation, versus overall chronological age?
We agree with the reviewer that it would be interesting to investigate how a scar develops in a young child versus an older child. However, this was beyond the scope of this particular validation study. Nevertheless, we have added this suggestion to the Discussion section, line 230-231.
Material and methods. This section is well described.
Results:
This is section is well detailed. The authors may want to include some further statistical data into Table 1, to evaluate for possible differences between the groups.
The 46 children who filled out the SCAR-Q, are a subgroup of the 100 children who were evaluated by the SBSES. Since this per definition leads to a subgroup bias, we did not analyses possible differences between the groups. Since we did not collect data from non-participants, a comparison between those groups was not possible either, as we mentioned in the Discussion section, line 227-228.
Introduction and discussion: both sections are well written and documented. The authors may consider to discuss some of the limitation of a single center study and also the impact of other treatment onto scar formation and acceptance. Some patients having suffered from solid tumors and who are long term survivors may not be bothered as much by a scar compared to patients with benign disease? Could the authors comment on that. Radiation fields, other medical therapies, especially steroids may worsen healing processes and make scar look worse. Please comment.
Thank you for bringing up these additional points of discussion. We have added the limitation of a single center study in line 226-227. The surgical procedures included in this study have been outlined in Table 1. We did not include any solid tumors. The three children included underwent a fasciotomy, excision of a lipoma, and excision of an unspecified skin tumor. None of the children underwent additional therapy such as radiation or steroids. Nonetheless, we have Discussion these interesting topics on page 9, line 230-233.
Minor comment:
Some spelling errors and minor grammatical inaccuracies should be corrected. A few medical terms should be translated into English, i.e. hernia cicatricalis is likely incisional hernia, necrotic enterocolitis is usually described as necrotizing enterocolitis in the English literature.
We have corrected the spelling errors throughout the manuscript.
Round 2
Reviewer 3 Report
Comments and Suggestions for Authors
The final version can be accepted